# Myocardial and Vascular Involvement in Patients with Takayasu Arteritis: A Cardiovascular MRI Study

**DOI:** 10.3390/diagnostics13233575

**Published:** 2023-11-30

**Authors:** Simin Almasi, Sanaz Asadian, Leila Hosseini, Nahid Rezaeian, Shakiba Ghasemi asl, Abdolmohammad Ranjbar, Seyyed-Reza Sadat-Ebrahimi, Behnaz Mahmoodieh, Alireza Salmanipour

**Affiliations:** 1Department of Rheumatology, Firoozgar Hospital, Iran University of Medical Sciences, Tehran 1593747811, Iran; simin_almasi@yahoo.com; 2Rajaie Cardiovascular Medical and Research Center, Iran University of Medical Sciences, Tehran 1995614331, Iran; asadian_s@yahoo.com (S.A.); hosseini1115@yahoo.com (L.H.); asalmanipour@gmail.com (A.S.); 3School of Medicine, Iran University of Medical Sciences, Tehran 1449614535, Iran; shaakiibaa@gmail.com; 4Cardiovascular Research Center, Madani Heart Hospital, Faculty of Medicine, Tabriz University of Medical Sciences, Tabriz 5166616471, Iran; dr.am.ranjbar@gmail.com (A.R.); rezasadat46@gmail.com (S.-R.S.-E.); 5Molecular and Experimental Cardiology Research Center, Ruhr University Bochum, 44803 Bochum, Germany; 6Young Researchers and Elite Club, Tehran Medical Sciences, Islamic Azad University, Tehran 193951495, Iran; behnaz_mahmoodieh@yahoo.com; 7Cardiovascular Imaging Research Center, Iran University of Medical Sciences, Tehran 1995614331, Iran

**Keywords:** Takayasu arteritis, cardiovascular MRI, strain, feature tracking, cardiac and vascular complications

## Abstract

We aimed to explore the cardiovascular magnetic resonance (CMR) of Takayasu arteritis (TA) and its cardiovascular complications. CMR was conducted on 37 TA patients and 28 healthy individuals. We evaluated the CMR findings and adverse cardiovascular complications at the time of the CMR (ACC_CMR_). After 8 to 26 months, the major adverse cardiac and cerebrovascular events (MACCEs) were evaluated. The TA included 25 women (67.6%), aged 36 ± 16 years old, and 28 age- and sex-matched healthy controls. Left ventricular (LV) ejection fraction was significantly lower in the TA group than in the control group (51 ± 9% vs. 58 ± 1.7%; *p* < 0.001). Aortic mural edema was present in 34 patients (92%) and aortic mural hyperenhancement in 36 (97%). Left ventricular global longitudinal strain (LVGLS) was significantly lower in the TA group (median [interquartile range] = 13.70 [3.27] vs. 18.08 [1.35]; *p* < 0.001). ACC_CMR_ was seen in 13 TA patients (35.1%), with the most common cardiac complication being myocarditis (16.2%). During a median follow-up of 18 months (8–26 months), nine patients developed MACCEs, of which the most common was cerebrovascular accident in five (13.5%). The LVGLS of the CMR had the strongest association with complications. Myocardial strain values, especially LVGLS, can reveal concurrent and future cardiovascular complications in TA patients.

## 1. Introduction

Takayasu arteritis (TA) is a chronic vasculitis involving large vessels, cardiac valves, myocardium, and coronary arteries. The estimated incidence rate of TA across the world ranges from 0.3 to 3.3 per million per year, and the prevalence ranges between 4.7 and 360 cases per million [1]. The accurate detection and prompt diagnosis of cardiac and vascular involvement in TA patients are vital for intervention within the potentially reversible stages. A retrospective multicenter study of 318 TA patients reported that approximately half of patients experienced relapse or vascular complications within 10 years from diagnosis, leading to significant morbidity and mortality, particularly in severe cases. Previously, X-ray angiography was assumed to be the gold standard in detecting vascular impairment and diagnosing TA. Nonetheless, recently, cardiovascular magnetic resonance (CMR) imaging, on the strength of its potential for presenting high-resolution imaging of thickening, luminal alterations, and aneurysm formation in large- and medium-sized vessels, has attracted particular interest in the evaluation of TA patients [2]. Aside from its advantages as a noninvasive and radiation-free procedure compared with invasive angiography, CMR helps detect myocarditis in TA patients. CMR tissue characterization assists in the early identification of cardiovascular complications in a variety of autoimmune rheumatological disorders and contributes to patient management. The cine function depicts cardiac anatomy and its function. The T2-weighted (W) and the newer T2-mapping sequences detect myocardial edema. The late gadolinium-enhanced T1-W images demonstrate myocardial replacement with fibrosis, and the T1 mapping and extracellular volume fraction (ECV) assist in quantifying diffuse myocardial fibrosis [3].

Several CMR techniques can measure cardiac muscle motion and structure. Strain imaging using feature-tracking (FT) CMR offers fresh insight into the severity and progression of various cardiac diseases. Several studies have reported a correlation between both global and segmental myocardial strain values and the amount of late gadolinium enhancement (LGE) and prognosis [4,5,6,7,8,9].

Therefore, several studies have posited that strain analysis using FT-CMR might serve as a diagnostic modality and a risk-assessment and treatment-tailoring tool in various inflammatory diseases with cardiac involvement. In the current study, we aimed to describe the CMR characteristics of TA patients and compare them with the CMR characteristics of healthy control counterparts. Moreover, we investigated the role of CMR parameters in predicting concurrent and future major adverse cardiac and cerebrovascular events (MACCEs).

## 2. Material and Methods

### 2.1. Study Population

This prospective study was conducted as per the Helsinki Declaration and its later amendments, and the study protocol was approved by our institutional ethics committee. Written informed consent was obtained from each patient. All patients with a definite diagnosis of TA, as established by an expert rheumatologist based on laboratory tests, clinical signs and symptoms, and imaging findings, were consecutively included from January 2018 through January 2021. All TA patients fulfilled the diagnostic criteria for TA following the diagnostic guidelines of the American College of Rheumatology in 1990 [10]. A control group was recruited from age- and sex-matched healthy volunteers. The exclusion criteria were contraindications to CMR, having implanted cardiac devices (e.g., pacemakers, implantable cardioverter-defibrillators), glomerular filtration rates (GFRs) below 30 mL/m^2^, and significant cardiac arrhythmias resulting in poor image quality.

The demographic characteristics, medical history, and the result of the erythrocyte sedimentation rate (ESR) test were recorded. The signs and symptoms of the TA patients were registered in a preprepared checklist. The entire study population underwent CMR for the first time.

### 2.2. Cardiac MRI Imaging

The CMR studies were performed utilizing a 1.5-Tesla scanner (MAGNETOM Avanto, Siemens Healthcare, Erlangen, Germany), featuring an 8-element cardiac-phased array receiver surface coil with the patient in the supine position.

Functional electrocardiography-gated sequences were acquired during an end-expiratory breath hold.

Cardiac size and systolic function were assessed in a stack of contiguous cine short-axis and horizontal and vertical long-axis slices (slice thickness = 8 mm, field of view = 300 mm, imaging matrix = 156 × 192, no interslice gaps in the short-axis stacks, and repetition time/echo time = 31/1.2 ms). Ventricular size and volume were determined by tracing the end-diastolic and end-systolic endocardial borders on the cine short-axis images.

Myocardial edema was diagnosed via short tau inversion recovery (STIR) sequences as an area of a myocardial-to-skeletal muscle signal intensity ratio exceeding 1.9.

Hyperemia was assessed by pre- and early (within 2 min) post-contrast T1-W images as a myocardial-to-skeletal muscle signal intensity ratio of greater than 4. Myocardial fibrosis was evaluated visually using LGE sequences.

Furthermore, the aorta was evaluated in the candy-cane view utilizing cine functional, STIR, and LGE sequences (Figure 1) An electrocardiogram-gated double-inversion recovery black-blood fast spin-echo image was acquired to calculate the thickness of the aortic wall (field of view = 36 × 36 cm, repetition time = 2 R-R intervals, time to echo = 40 ms, slice thickness = 5 mm, and matrix size = 512 × 256). Axial images were obtained at the right pulmonary artery level to measure the descending thoracic aorta. The normal aortic wall thickness was defined based on a multi-ethnic study on atherosclerosis [11].

Major aortic thoracoabdominal branch involvement was recorded by assessing aortic arch subdivisions, including brachiocephalic, bilateral subclavian, and bilateral common carotid arteries, as well as abdominal aorta branches, consisting of the celiac trunk, the superior mesenteric artery, and the bilateral renal arteries. The number of affected major branches was reported.

The FT data were analyzed offline using the CVI 42 software (Circle Cardiovascular Imaging, Calgary, AB, Canada), version 5.6.2. The endocardial and epicardial borders were manually traced in the end-diastolic frame and then propagated through the cardiac cycle.

Three-dimensional left ventricular (LV) strains were extracted from the 2-, 3-, and 4-chamber planes, as well as the short-axis stack, to characterize the global longitudinal strain (GLS), global circumferential strain (GCS), and global radial strain (GRS) (Figure 2). All strain values were expressed as absolute values.

Additionally, the adverse cardiovascular complications at the time of the CMR (ACC_CMR_) were determined in order to show any differences among the TA patients with or without these complications. The disorders assessed were as follows:Dilated cardiomyopathy (DCM): defined based on the accepted literature [12,13];Myocarditis: defined based on the appropriate clinical, laboratory, and imaging findings [12,13];Severe aortic regurgitation: defined based on the guidelines regarding valvular heart diseases [14];Aortic aneurysms and aortic dissection: extracted from the patient’s computed tomography angiography (CTA) reports;Main pulmonary artery aneurysms: registered based on the patient’s CTA and CMR findings.

### 2.3. Major Adverse Cardiac and Cerebrovascular Events at Follow-Up

The patients were followed up for between 8 and 26 months (median = 18 months) for the occurrence of any cardiac or vascular complications of TA by assessing medical records and telephone calls.

The MACCE at the time of the follow-up was defined as:Death: based on a telephone follow-up of the patient;Myocardial infarction (MI): based on the patient’s telephone follow-up and medical records;Aortic dissection: based on the patient’s aortic CTA results;Pulmonary thromboembolism: diagnosed based on the patient’s pulmonary CTA results;Cerebrovascular accident (CVA) and transient ischemic attack (TIA): diagnosed according to the diagnostic guideline of the American Academy of Neurology [15];Coronary artery bypass graft surgery: based on the availability of the patient’s surgical records and files.

### 2.4. Disease Activity Criteria and Corticosteroid Use

Disease activity was determined based on clinical and lab tests, such as fever, arthritis, high ESR and C-reactive protein (CRP) values, and CMR findings of aortic wall or branch edema. In terms of corticosteroid use, consumption of less than 15 mg/day was considered a low dose, and injection of 1 mg/kg or pulse methylprednisolone was considered a high-dose corticosteroid treatment. The main study endpoint was to evaluate the association between CMR parameters and adverse event in patients with Takayasu arteritis.

### 2.5. Statistical Analysis

The quantitative variables are reported as the mean ± standard deviation (SD), and the qualitative variables are reported as the frequency (percentage). The normal distribution of all values was tested using the Shapiro–Wilk test. The independent samples *t*-test was utilized to compare the quantitative measures with a normal distribution and the Mann–Whitney test for the nonparametric variables between the groups. The chi-squared or Fisher exact test was applied for the qualitative parameters. The statistical analyses were conducted using the SPSS software, version 24 (SPSS Co., Chicago, IL, USA). Two-sided tests were conducted with a significance level at 0.05.

## 3. Results

In the present study, 37 TA patients and 28 age-matched healthy controls were evaluated. The baseline characteristics of the participants in both groups and the complications in the TA group are described in Table 1. The majority of the TA patients were female, 25 (67.6%), and they had a mean ± SD age of 36 ± 16 years. There were no statistical differences between the two groups concerning sex (*p =* 0.1) and age (*p =* 0.4). The most common signs and symptoms of our TA patients at presentation were dyspnea (56.7%), chest pain (37.8%), fatigue (37.8%), upper-extremity blood pressure difference (32.4%), palpitations (16.2%), and visual field defects (10.8%).

### 3.1. CMR Findings

Left ventricular ejection fraction (EF) was significantly lower in the TA group than in the control group (51 ± 9% vs. 58 ± 1%; *p* < 0.001). The frequencies of the affected major arteries are demonstrated in Figure 3. No major aortic branch was involved in 13 patients (35.1%). One or two major branches were affected in 14 patients (37.8%), with 7 patients (18.9%) in each category. Three or more major aortic branches were involved in 10 patients (27%). Aortic mural edema, evidenced by a high signal intensity in the STIR sequences, was present in 34 patients (92%), and aortic mural hyperenhancement was present in 36 patients (97%) as well.

Absolute LVGLS was significantly lower in the TA group than in the control group (mean [SD] = 13.70 [3.27] vs. 18.08 [1.35]; *p* < 0.001). LVGCS and LVGRS were not significantly different between the two groups (*p =* 0.1 and *p =* 0.3, respectively).

There was no significant correlation between the LV strain parameters and the number of affected arteries (*p* > 0.05 for all parameters).

Abnormal cardiovascular complications at the time of CMR (ACC_CMR_) were noted in 13 patients (35.1%): myocarditis in 6 patients (16.2%), dilated cardiomyopathy in 1 (2.7%), severe aortic regurgitation in 2 (5.4%), main pulmonary artery aneurysm in 1 (2.7%), left pulmonary artery stenosis in 1 (2.7%), aortic dissection in 1 (2.7%), and ascending aorta aneurysm in 1 (2.7%) (Table 1).

The results of the independent samples *t*-test for the comparison of LV functional parameters (viz, LVEF, LVGLS, LVGCS, and LVGRS) between uncomplicated TA patients versus those with ACC_CMR_ and MACCEs are depicted in Table 2.

### 3.2. Clinical, Laboratory, and Treatment Data

Thirty-four patients (92.8%) received corticosteroids. The corticosteroid dose was divided into two categories: low dose (<15 mg/day) and high dose (>1 mg/kg or methylprednisolone pulse) according to our rheumatologist’s prescription. Twenty-eight out of the 34 patients (82%) received high-dose corticosteroids, and 6 out of the 34 patients (17.6%) received low-dose corticosteroids.

According to the laboratory analysis, only eight patients (21%) had high levels of ESR. In the entire study population, in addition to steroids, other drugs such as methotrexate, azathioprine, mycophenolate, cyclophosphamide, and anti-TNF drugs were used. Twenty-one patients (57%) received cyclophosphamide, four (10.7%) received mycophenolate, and nineteen (50%) received anti-TNF drugs.

Six months after the initial treatment and the gradual tapering of the steroid dose, if the disease was in remission (based on the aforementioned disease-activity criteria), methotrexate or azathioprine was continued. Additionally, anti-TNF was replaced with anti-IL-6 if the disease was resistant to treatment. Among the 19 patients who received anti-TNF, 14 patients received the drug from the outset, and 5 patients were switched to anti-IL-6.

Only one patient had resistance to anti-TNF treatment and was treated with tocilizumab, which decreased the inflammatory factors significantly.

### 3.3. Follow-Up Results

During a follow-up period of 8 to 26 months (median = 18 months), nine patients developed MACCEs: CVA in five patients (13.5%), pulmonary thromboembolism in one (2.7%), TIA in one (2.7%), and coronary artery disease necessitating coronary artery bypass graft surgery in two (5.4%) (Table 1). The disease complications among the group with MACCEs (nine patients) occurred from 3 to 6 months after CMR examination.

Comparisons of the CMR’s functional parameters between the two groups with and without MACCEs during the follow-up are demonstrated in Table 2. LVGLS was the only parameter with a significant difference (*p* = 0.02). Comparisons of the patients with and without MACCEs concerning ESR during the follow-up demonstrated mean ESR values of 35.17 ± 29.72 and 29.25 ± 33.29, respectively (*p* = 0.6). The mean ESR values were 27.84 ± 26.69 and 42.83 ± 34.28 (*p* = 0.1) in patients with and without complications at the time of CMR, respectively.

Among the TA patients receiving high-dose corticosteroids, eight patients (28%) developed MACCEs, while in six patients on low-dose corticosteroids, MACCEs were reported in one patient (17%).

## 4. Discussion

TA is a granulomatosis vasculitis that predominantly involves the aorta, the medium-sized aortic branches, and the pulmonary arteries. Cardiac involvement has been reported previously in different types of vasculitis, including TA and other inflammatory disorders [16]. To our knowledge, there is limited information regarding MRI features especially advanced modalities including FT, in TA patients within the available literature. Therefore, in the present study, we primarily sought to explore CMR findings concerning ventricular function, myocardial strains, and tissue characterization, and involvement of the aorta, and its branches. Additionally, we compared all of these parameters with those in age- and sex-matched healthy individuals. Moreover, we stratified our TA patients according to the presence or absence of ACC_CMR_ and MACCE during the follow-up to search for a potential association of ACC_CMR_ and MACCE with the imaging findings.

We found several compelling and noteworthy results in this study that may impact the clinical management of TA patients. Our study revealed that the most affected major aortic branches were the subclavian arteries, predominantly the left subclavian artery. Furthermore, approximately half of our TA patients had involvement in at least two major aortic branches at the time of presentation, underscoring the extent to which TA could affect major aortic branches. Aortic mural thickening and edema were frequent among the TA group. About 5% of our cases also demonstrated pulmonary artery involvement. We demonstrated that both LVEF and LVGLS measured using CMR were significantly lower in the TA group than in the healthy control group. All three global LV strain values were significantly lower in the TA patients with ACC_CMR_ compared to the TA patients without such complications. Interestingly, about 24% of the TA patients developed MACCEs during the follow-up period although most of these events occurred around 3 to 6 months after CMR examination. LVGLS was the single different functional parameter between the TA patients with and without MACCEs. Nevertheless, the mean ESR level or the corticosteroid dose was not statistically different between the MACCE-positive or MACCE-negative groups during the follow-up.

Using CMR sequences, we evaluated the entire thoracic and abdominal aorta and their branches regarding vessel-wall thickening, inflammation, and fibrosis and found that the subclavian artery, followed by the common carotid artery, was the most commonly affected aortic branches. In line with our findings, in a study conducted by Soriano et al. [17], which employed positron emission tomography (PET) for vascular assessment, the left and right subclavian arteries and the left and right common carotid arteries had the highest maximum standardized uptake value, suggesting higher arterial involvement in the mentioned vascular districts. Additionally, the study by Park et al. [18], employing CTA, magnetic resonance angiography, and conventional arteriography, aligned with our study, reported that the left subclavian artery had the highest rate of occlusion or stenosis among aortic branches; however, the right common carotid artery involvement rate was nearly two-fold that of the right subclavian artery. In our investigation, two-thirds of TA patients demonstrated aortic arch branch involvement, while abdominal aortic branches were affected in less than one-third of the patients. It seems that the pattern of vessel involvement in TA has significant ethnic differences. Hata et al. demonstrated that the aortic arch and its branches are particularly affected in Japanese patients, while the renal and abdominal aorta are mainly involved in the Indian population [19]. Additionally, there are some reports regarding a decrease in the aortic and pulmonary wall thickness following steroid therapy utilizing MRA [20].

All our patients exhibited aortic mural thickening, and a majority showed evidence of mural inflammation, including edema, which confirms the ability of CMR to evaluate vascular thickening and vessel-wall inflammation. Andrews et al. demonstrated the superiority of MRI over X-ray angiography in early diagnosis and therapy guidance [21]. Despite the capability of MRI to reveal vessel wall inflammation, the correlation between MRI findings and disease activity is controversial [22,23].

Two of the TA patients had pulmonary vessel involvement: aneurysmal dilation of the main pulmonary artery and its branches in one patient and left pulmonary artery inflammation and stenosis in the other one (Figure 4). Yamada et al. [24] demonstrated that CMR, compared with invasive angiography, could correctly display 50% pulmonary artery involvement in TA patients. Similar to our investigation, in a multicenter computed tomography angiography (CTA) study conducted by Yang et al. [25], among 815 patients diagnosed with TA, 51 patients (6%) had evidence of pulmonary arteritis, of which 82% of patients had pulmonary parenchymal lesions on CT scan, and 59% had pulmonary hypertension (PH) on echocardiography. In a study of 126 patients with TA, the pulmonary artery was affected in 15% of their subjects, and it had correlation with more prolonged disease duration and more symptoms of hemoptysis. They found that pulmonary arteries typically display stenosis and occlusion rather than aneurysm [26].

In terms of cardiac function, LVEF and LVGLS were both impaired in patients with TA compared with the healthy controls. The precise underlying mechanism of LV deterioration in TA has yet to be determined. As a plausible hypothesis, the disturbance in the elastic deformation of the cardiac walls could potentially be attributed to the extensive inflammatory reaction [27]. Moreover, these cardiac changes may be a mechanical response to the increased arterial stiffness in TA patients [28]. Various factors, including arterial hypertension, aortic regurgitation, coronary artery disease, pulmonary vascular, and autoimmune mechanisms can lead to cardiac involvement [29]. While we could not find any significant association between the number of affected aortic branches and LV functional and strain parameters, it might be partly due to our small sample size.

We observed that 35% of our TA patients had abnormal cardiovascular findings at CMR. The most frequent conditions were myocarditis, identified in six patients, and severe aortic regurgitation, observed in two cases. In line with our study, previous studies reported myocardial involvement in approximately 25 to 53% of TA patients in various studies [28,30,31]. It was suggested that myocardial involvement in TA patients might be associated with factors such as early age of disease onset, lack of classic risk factors, and more active disease [30]. Via traditional and novel parametric criteria, CMR also has a diagnostic role vis-à-vis vasculitis in the other large vessels, such as giant cell arteritis, and can reveal myocardial involvement, including myocarditis, in these patients [32]. Nonetheless, regarding TA, CMR is beneficial in detecting cardiac involvement, but its role in the diagnosis and assessment of the activity of the disease is of debate. The detected pattern of myocardial fibrosis in our study population was mostly mid-myocardial and subepicardial, reflecting the probable myocardial inflammation leading to fibrosis. Likewise Chandrashekhara et al. reported that the most common pattern of LGE in TA was mid-myocardial enhancement (13.8%), pursued by epicardial fibrosis in four (11.11%) patients [33].

In the absence of an overt difference in LVEF, all LV strain values were significantly lower in the TA patients with concurrent cardiovascular disorders than in those without such complications. Our findings underscore the role of comprehensive CMR in determining cardiovascular complications that may not be detectable through conventional measures such as LVEF. Aligned with our findings, in a previous study, the majority of TA patients exhibited decreased LVGLS and longitudinal early diastolic strain rates, even in subgroups of patients without LGE and subgroups of patients with LVEF above 55% [34]. Therefore, LV strain analysis, especially LVGLS, could be used as a sensitive marker of myocardial involvement in TA patients. We postulated that respecting the pattern of overt or subtle myocardial fibrosis in the mid- and subepicardial layer, LVGCS is impaired in most TA subjects regardless of the disease course. However, impairment of LVGLS reflects subendocardial deformation and possibly distinguishes TA with more severe complications. In accordance with our findings, Guo et al. [35] reported a decreased global and segmental myocardial strain using FT-CMR in TA patients with preserved LVEF compared to healthy controls. They also reported an association between PH, male gender, elevated ESR, and myocardial LGE with declined LV strains. Furthermore, they found a stronger association between reduced strains with PH rather than arterial hypertension; probably as a consequence of ventricular interdependence. Another study demonstrated that TA patients with reduced LVEF had significantly lower LVGLS compared to the ones with preserved LVEF. This further highlights the fact that LVGLS could serve as an assessment tool for cardiac involvement in TA patients both with and without reduced LVEF [34].

About one-quarter of our study subjects developed MACCEs during the follow-up period, with neurovascular events as the leading complication, comprising five cases of CVA and one case of TIA. According to prior investigations, CVA and TIA are common in TA patients, although their morbidity and mortality within this population need further elucidation. Vascular complications are mostly connected to thoracic aortic involvement, retinopathy, and the progressive nature of the disease [29]. The debilitating nature of the mentioned conditions mandates a measure for the risk estimation of cerebrovascular complications to take potential preventive measures.

Interestingly, our results revealed no significant difference in the mean ESR level between TA patients with and without MACCEs during the follow-up period, nor was there any significant association between corticosteroid dose (high-dose vs. low-dose) and the occurrence of MACCEs. A recent study conducted by Yu et al. [36] on 52 TA patients, reported that 15 (29%) of these patients experienced a severe ischemic event (ISE) within the 6-year follow-up period, of which 5 experienced CVA, 5 encountered acute coronary syndrome (ACS), 1 exhibited ischemic cardiomyopathy, and 5 experienced limb ischemia. They further investigated the potential predictors of ISE in TA patients. They demonstrated that TA patients receiving a combination of corticosteroids and another immunosuppressant agent are more likely to encounter ISE compared to the ones who do not receive it besides corticosteroids. However, they did not find any significant association between demographic or laboratory findings and the occurrence of ISE in patients with TA.

In this study, we attempted to portray the role of CMR in diagnosis and as a prognostic modality in patients with TA. Having the ability to detect myocardial and aortic branch vessel involvement makes it an outstanding imaging technique. Numerous adverse cardiovascular events can occur in TA that require an exact follow-up evaluation. Future large-scale cohort studies should determine the role of CMR in diagnosing and managing patients with TA.

## 5. Limitations

In the current study, the sample size, albeit acceptable, precluded subgroup analysis. The recruitment of patients with TA is bound to be hampered by the rarity of the disease. Thus, even though this study was conducted at a tertiary referral center, the sample size was not very large. Consequently, future larger-scale multicentric studies are warranted to validate our results. Among the MACCEs groups, most of the adverse events occurred 3 to 6 months after the CMR examination; therefore, an analysis of the CMR predictors with the Cox regression model was not possible. A longer follow-up period with a larger cohort is warranted [37]. In the current investigation, electronic records and telephone calls were utilized to assess MACCEs; therefore, clinically diagnosed complications such as heart failure could not be reported. It is also worth noting that T2W STIR sequences are associated with inherent vulnerability to artifacts and may, thus, pose interpretation challenges. Lastly, strain parameters are affected by different variables including the presence of any myocardial damage, which requires subgroup analysis in a far larger study population [38].

## 6. Conclusions

CMR and FT-CMR could assist in the assessment of both the heart and all major arteries in TA patients in a single imaging session, which holds the potential for the early identification of cardiovascular involvement, even in advance of LVEF reduction. CMR-derived deformation values, specifically LVGLS, might be an indicator of the presence of ACC_CMR_ or the future occurrence of MACCEs during follow-ups. Therefore, CMR and FT-CMR could play a potential role in the early detection of cardiovascular involvement and cardiovascular risk stratification in TA patients.

## Figures and Tables

**Figure 1 diagnostics-13-03575-f001:**
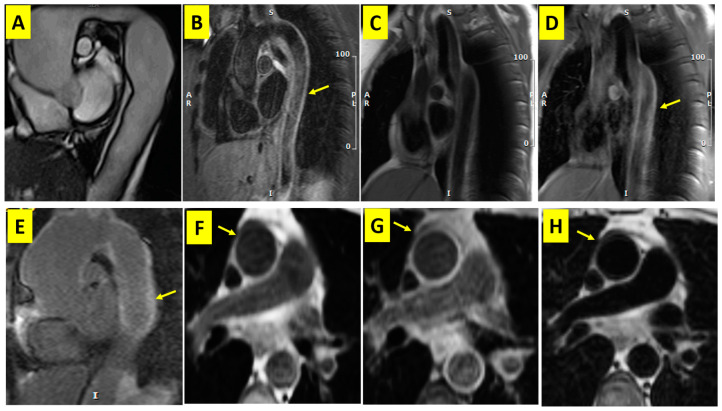
Cardiac MRI sequences of Takayasu arteritis from ascending, arch, and descending thoracic aorta in the candy-cane view and images in a healthy control subject: (**A**) cine function showing dilation and irregularity in the size of the aorta; (**B**) STIR image depicting descending aorta mural inflammation (yellow arrow); (**C**,**D**) pre- and post-contrast T1-weighted images showing increased gadolinium uptake after contrast injection (yellow arrow); (**E**) LGE image demonstrating increased aortic wall gadolinium uptake (yellow arrow); (**F**) axial T1-W black-blood fast spin-echo; (**G**) axial T1-W black-blood fast spin-echo with fat saturation; (**H**) axial T2-W black-blood fast spin-echo demonstrating normal aortic wall thickness (yellow arrow) in a healthy control person. STIR: short tau inversion recovery; LGE: late gadolinium enhancement.

**Figure 2 diagnostics-13-03575-f002:**
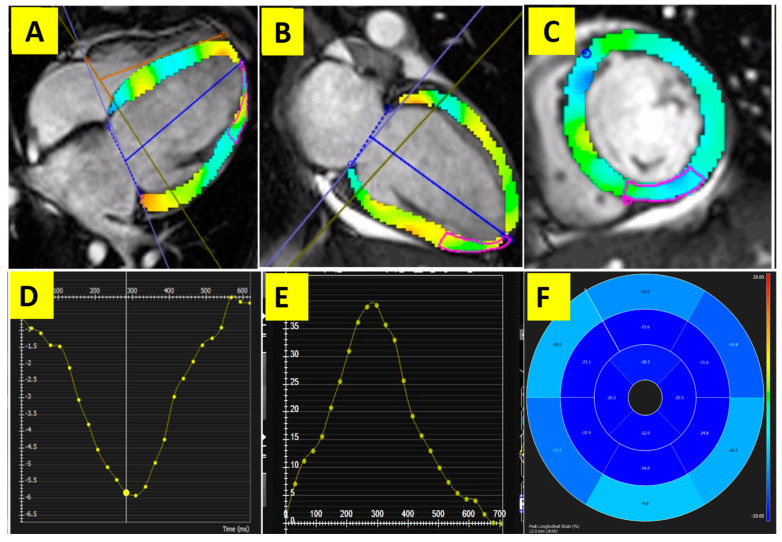
Feature tracking of the LV strain: (**A**–**C**) 2-chamber, 4-chamber, and LV short-axis LV wall (colorized) to derive the strain, respectively; (**D**,**E**) longitudinal and radial strain curves, respectively; (**F**) longitudinal polar map depicting the segmental strains, colorbar shows strain range.

**Figure 3 diagnostics-13-03575-f003:**
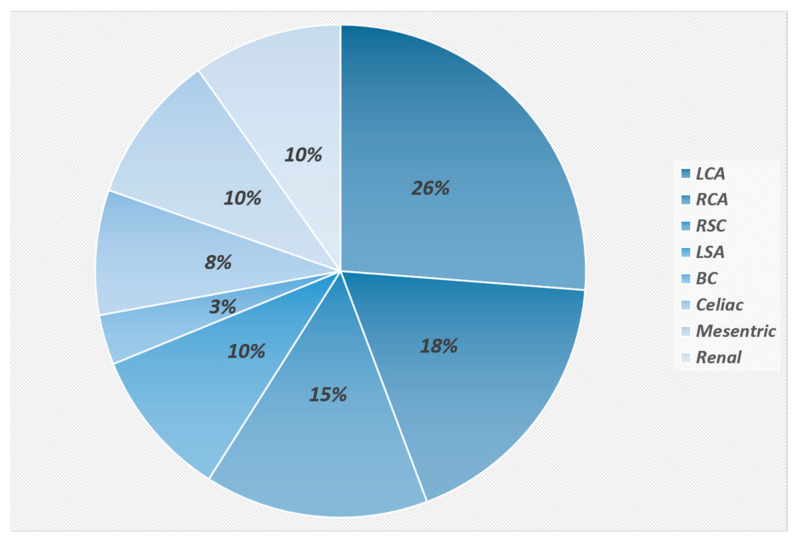
Distribution of the affected arteries in patients with Takayasu arteritis. LCA and RCA, left and right common carotid artery; RSCand LSA, right and left subclavian artery; BC, brachiocephalic artery; mesenteric, compatible with superior mesenteric artery.

**Figure 4 diagnostics-13-03575-f004:**
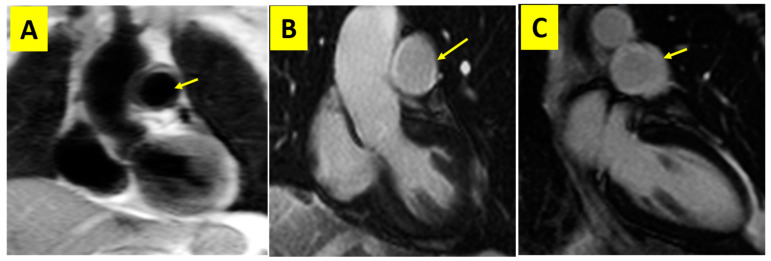
(**A**) Black-blood localizer image showing increased wall thickening of the main pulmonary artery (yellow arrow); (**B**,**C**) LGE images depicting gadolinium enhancement of the main pulmonary artery compatible with pulmonary arteritis (yellow arrow). LGE, late gadolinium enhancement.

**Table 1 diagnostics-13-03575-t001:** Baseline and follow-up characteristics of the study participants.

		Patient (*n* = 37)	Control (*n* = 28)	*p*-Value
Male		12 (32.4)	14 (50.0)	0.1
Female		25 (67.6)	14 (50.0)	
Age *y* mean (SD)		36 (16)	31 (4)	0.4
Hypertension *n* (%)		6 (16.2)	0 (0)	-
Diabetes *n* (%)		1 (2.7)	0 (0)	-
Smoking		2 (5.4)	0 (0)	-
Dyslipidemia		3 (8)	0 (0)	-
ESR mean (SD) ^†^		39 (30)	3 (1)	<0.001
LVEF % mean (SD) ^†^		51 (9)	58 (1.7)	<0.001
Strain Parameters mean (SD)				
LVGLS ^†^		13.70 (3.27)	18.08 (1.35)	<0.001
LVGCS		17.44 (3.95)	18.69 (2.18)	0.1
LVGRS		42.54 (15.37)	39.83 (7.78)	0.3
General characteristics of the Takayasu arteritis patients				
Characteristics	Variable	Number	Variable	Number
Signs and Symptoms at Presentation *n* (%)	Dyspnea	21 (56.7)	Chest pain	14 (37.8)
	Fatigue	14 (37.8)	Upper-extremity blood pressure difference	12 (32.4)
	Palpitations	6 (16.2)	Visual field defects	4 (10.8)
Number of the Affected Major Aortic Branches at Presentation *n* (%)	No involvement	13 (35.1)	1 Affected vessel	7 (18.9)
	2 Affected vessels	7 (18.9)	≥3 Affected vessels	10 (27)
Affected Major Aortic Branches at Presentation (*n* %)	Left subclavian	16 (43.2)	Right subclavian	11 (29.7)
	Left common carotid	9 (24.3)	Right common carotid	6 (16.2)
	Superior mesenteric	6 (16.2)	Left or right renal	6 (16.2)
	Celiac trunk	5 (13.5)	Brachiocephalic	2 (5.4)
Presence of Aortic Wall Thickening	Thickened	37 (100)	Normal	0
Aortic Mural Edema * at Presentation *n* (%)	Positive	34 (92)	Negative	4 (11)
Aortic Hyperenhancement ^¥^ at Presentation *n* (%)	Positive	36 (97.3)	Negative	1 (2.7)
Myocardial Subepicardial and Mid-Wall Fibrosis ^¥^ at Presentation *n* (%)	Positive	6 (16.3)	Negative	31 (83.7)
ACC_CMR_ (%)	DCM	1 (2.7)	Main PA aneurysm	1 (2.7)
	LPA stenosis	1 (2.7)	Myocarditis	6 (16.2)
	Severe AR	2 (5.4)	Aortic dissection	1 (2.7)
	Ascending aorta aneurysm	1 (2.7)		
MACCE at Follow-Up *n* (%)	CVA	5 (13.5)	PTE	1 (2.7)
	TIA	1 (2.7)	CABG	2 (5.4)

ESR, erythrocyte sedimentation rate; LVEF, left ventricular ejection fraction; CVA, cerebrovascular accident; TIA, transient ischemic attack; PTE, pulmonary thromboembolism; DCM, dilated cardiomyopathy; CABG, coronary artery bypass graft surgery; AR, aortic valve regurgitation; LVGLS, left ventricular global longitudinal strain; LVGCS, left ventricular global circumferential strain; LVGRS, left ventricular global radial strain; ACC_CMR_, adverse cardiovascular complications at the time of CMR; MACCE, major adverse cardiac and cerebrovascular events. * At short TI inversion recovery (SITR) sequence. ^†^ Variables with nonparametric distribution. ^¥^ Late gadolinium enhancement.

**Table 2 diagnostics-13-03575-t002:** Comparison of LV functional parameters, age, and ESR between uncomplicated TA patients versus TA with ACC_CMR_ or MACCE.

	ACC_CMR_ (at the Time of CMR)			MACCE (at Follow-Up)		
Variable	Mean ± SD with ACC_CMR_	Mean ± SD without ACC_CMR_	*p*	Mean ± SD with MACCE	Mean ± SD without MACCE	*p*
**LVEF**	47.72 ± 11.49	52.52 ± 7.05	0.1	49.11 ± 12.22	51.39 ± 7.92	0.5
**LVGLS**	12.08 ± 3.54	14.57 ± 2.82	0.02	11.61 ± 3.81	14.37 ± 2.84	0.02
**LVGCS**	15.47 ± 4.10	18.51 ± 3.50	0.02	15.55 ± 5.79	18.05 ± 3.04	0.2
**LVGRS**	35.9 ± 15.51	46.12 ± 14.36	0.05	37.19 ± 20.58	44.26 ± 13.30	0.2
**Age**	35.85 ± 13.78	36.29 ± 17.52	0.9	42.89 ± 9.35	33.97 ± 17.32	0.1
**ESR**	42.83 ± 34.28	27.84 ± 26.69	0.1	29.25 ± 33.29	35.17 ± 29.72	0.6

CMR, cardiac magnetic resonance; MACCEs, major adverse cardiac and cerebrovascular events; ACC_CMR_, adverse cardiovascular complications at the time of CMR; LVGLS, left ventricular global longitudinal strain; LVGCS, left ventricular global circumferential strain; LVGRS, left ventricular global radial strain; LVEF, left ventricular ejection fraction.

## Data Availability

The data presented in this study are available upon request from the corresponding author. The data are not publicly available because of ethical considerations.

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
