# Peer review of "Myocardial and Vascular Involvement in Patients with Takayasu Arteritis: A Cardiovascular MRI Study"

_diagnostics, 2023, doi:10.3390/diagnostics13233575_

Round 1
Reviewer 1 Report
Comments and Suggestions for Authors
In my opinion, the main drawback of the study is its small number of patients, with a significant impact on the results. Notwithstanding this drawback, the data concern a rather rare pathology and, as regarding its cardiac implications, the findings of the study may have implications in clinical practice. It would have been interesting to see if echocardiographic exam corelate with CMR findings and clinical outcomes.
Author Response
Thanks for your concise and precise comments.
As you exactly mentioned, Takayasu arteritis is a relatively rare disease. We think that our result although in a small sample size can have management implications in this group of patients. However, our study and patient gathering are still continuing and we will report their follow-up in a larger population in the near future.
We are a tertiary center of cardiac imaging, with many referral patients. Many of our study population had echocardiography report which was done by different cardiologists with diverse qualities. Therefore, we decided to only report the ejection fraction data from the echocardiographic reports.
Reviewer 2 Report
Comments and Suggestions for Authors
Almasi Simin and colleagues performed an analysis on patients affected by Takayasu arteritis (TA) compared to healthy control individuals in order to investigate the role of cardiac magnetic resonance (CMR) findings in predicting concurrent and future major adverse cardiac and cerebrovascular events (MACCEs). 37 TA patients and a control group of 28 healthy individuals were included. Adverse cardiovascular complications at CMR (ACC-CMR) were seen in 13 TA patients (35.1%). During a median follow-up of 18 months (8–26 months), 9 patients developed MACCEs. LVGLS was the only independent predictor of MACCEs.
Some issues need to be addressed:
1) I have a few concerns regarding the statistical analyses:
i. First, all the continuous variables exhibited a normal distribution. This is somewhat surprising given the relatively small sample size. I'm curious about why the analysis used the Kolmogorav-Smirnov test instead of the Shapiro-Wilk test, which is typically more suitable for small sample sizes.
ii. Second, in cases where a non-normal distribution is detected, it would be more appropriate to use the Mann-Whitney test rather than the t-test.
iii. Lastly, I'm interested in the criteria used to select covariates for the logistic regression analysis. Furthermore, when it comes to predicting events at follow-up, a Cox regression would typically be the preferred method over logistic regression.
2) The study endpoints are not clearly outlined. We suggest adding a dedicated paragraph in the “Material and Methods” section.
3) Consider exploring the difference in FT (GLS etc) between patients with and without MAACEs. This approach could be more intriguing and informative than a straightforward comparison with the control group.
4) In the “Disease Activity Criteria and Corticosteroid Use” section, page 14, you stated that “consumption of less than 15 mg/day was considered a low dose, and injection of 1mg/kg or pulse methylprednisolone was considered a high-dose, corticosteroid treatment”. What about intermediate doses?
5) In the “Material and Methods” section you carefully described what you had considered as “MACCEs” and as “ACC-CMR”. Please consider adding some work from literature to explain why you picked those features as typical Takayasu arteritis’ markers.
6) Please specify this sentence: “Only 8 patients (21%) had high inflammatory factors in their blood.”
7) Please replace the fullstop with a comma at the end of line 13 of “Major adverse cardiac and cerebrovascular events at Follow-up” paragraph in the “Material and Methods” section.
8) The “Discussion” section is quite repetitive and redundant. Please consider making it more direct and concise. Moreover, consider also that GLS alterations could be a precursor of other diseases (e.g CAD/COVID alterations etc, 10.1111/echo.15431, 10.1093/ehjci/jead046). In addition, consider that LGE is a clear predictor in patients with myocardial damage (eg. 10.1016/j.jcmg.2023.05.016)
9) Table 1 layout could be improved. Please consider apport some minor changes to make it clearer.
Comments on the Quality of English LanguageRevision of English and checking abbreviations is recommended.
Author Response
I have a few concerns regarding the statistical analyses:
- First, all the continuous variables exhibited a normal distribution. This is somewhat surprising given the relatively small sample size. I'm curious about why the analysis used the Kolmogorav-Smirnov test instead of the Shapiro-Wilk test, which is typically more suitable for small sample sizes.
- Second, in cases where a non-normal distribution is detected, it would be more appropriate to use the Mann-Whitney test rather than the t-test.
Thanks for your great comment. We re-evaluated the variables and used the Shapiro-Wilk test.
LVEF, LVGCS, and ESR did not have normal distribution so we re-tested the data by Mann-Whitney test
- Lastly, I'm interested in the criteria used to select covariates for the logistic regression analysis. Furthermore, when it comes to predicting events at follow-up, a Cox regression would typically be the preferred method over logistic regression.
Thank you for your excellent comment. We think because of the chronic and ongoing inflammation some patients with TA are more at risk and show MACCE in follow-up. We aimed to evaluate any associations between CMR parameters and the occurrence of MACCE. Therefore, since the adverse effects were mostly around a few months from the CMR exam we prefer to analyze the data using logistic regression.
2) The study endpoints are not clearly outlined. We suggest adding a dedicated paragraph in the “Material and Methods” section.
Thanks for your clear comment. We added an endpoint to the material and method section.
3) Consider exploring the difference in FT (GLS etc) between patients with and without MAACEs. This approach could be more intriguing and informative than a straightforward comparison with the control group.
Thank you for your excellent comment. The second column of Table 2 is the comparison between FT parameters in patients with and without MAACEs and we highlighted it.
4) In the “Disease Activity Criteria and Corticosteroid Use” section, page 14, you stated that “consumption of less than 15 mg/day was considered a low dose, and injection of 1mg/kg or pulse methylprednisolone was considered a high-dose, corticosteroid treatment”. What about intermediate doses?
Thanks for your great comment. Our patients were referred from a rheumatologic center for a CMR exam. According to the dosage of corticosteroids to these patients and after consult with their rheumatologist we divided our patients into 2 groups.
5) In the “Material and Methods” section you carefully described what you had considered as “MACCEs” and as “ACC-CMR”. Please consider adding some work from the literature to explain why you picked those features as typical Takayasu arteritis markers.
Thank you very much for your comment. We have chosen the MACCEs and ACCCMR based on our patient's adverse events from the electronic report and also telephone call, although our literature review showed closely related results like Joseph G, Goel R, Thomson VS, Joseph E, Danda D. Takayasu Arteritis: JACC Focus Seminar 3/4. Journal of the American College of Cardiology. 2023 Jan 17;81(2):172-86.
6) Please specify this sentence: “Only 8 patients (21%) had high inflammatory factors in their blood.”
Thanks for your great comment. We mean that only * patients had high ESR above normal level and therefore we correct this sentence.
7) Please replace the full stop with a comma at the end of line 13 of the “Major adverse cardiac and cerebrovascular events at Follow-up” paragraph in the “Material and Methods” section.
Thanks for your smart consideration. We replaced full stops with commas in line 13 of the method and materials.
8) The “Discussion” section is quite repetitive and redundant. Please consider making it more direct and concise. Moreover, consider also that GLS alterations could be a precursor of other diseases (e.g CAD/COVID alterations etc, 10.1111/echo.15431, 10.1093/ehjci/jead046). In addition, consider that LGE is a clear predictor in patients with myocardial damage (eg. 10.1016/j.jcmg.2023.05.016)
Thank you very much for this meticulous viewpoint. We read the discussion again and tried to improve it as far as we could. We also added the limitations of the strain.
9) Table 1 layout could be improved. Please consider apport some minor changes to make it clearer.
Thanks for this great comment. We change the layout of the table 1.
Reviewer 3 Report
Comments and Suggestions for Authors
The current study evaluates cardiovascular aspects in patients with Takayasu Arteritis by CMR and potential prognosis markers derived from CMR and LV strain. It is a well conducted study and the paper is clearly written. Indeed, it is hard to make a bigger study as the pathology is rather rare. The findings are useful for clinical practice and the authors deserve appreciation. Minor comments:
- - Maybe reduce the results data from the discussion section, especially from the second paragraph
- - The title is a bit tricky as “myocardial mechanism” is a bit confusing after reading the whole article.
- - Revise the grammar and semantics.
Comments on the Quality of English Languageminor revision.
Author Response
Thanks for your concise and precise comments.
-Maybe reduce the results data from the discussion section, especially from the second paragraph
Thank you very much for your valuable comment. We summarized the important paragraph in the second paragraph of the discussion.
- - The title is a bit tricky as “myocardial mechanism” is a bit confusing after reading the whole article.
Thanks for this exact comment. We changed the title as follows: Myocardial and Vascular Involvement in Patients with Takayasu Arteritis: A Cardiovascular MRI study
- - Revise the grammar and semantics.
Thank you for your comment we have changed our manuscript grammar and semantics.
Round 2
Reviewer 2 Report
Comments and Suggestions for Authors
I thank the authors for the numerous responses and the great work done. Some points still need to be defined:
The response to the statistic issue is unsatisfactory as the median follow-up time is 18 months, as reported in the manuscript. I suggest trying a Cox regression instead of a logistic regression.
Finally, I recommend enhancing the discussion/limitation section by delving deeper into the GLS and LGE issues that were previously reported. It would be beneficial to expand the bibliography by including recent and relevant research articles that specifically address these topics. This will help provide a more robust and up-to-date foundation for readers.
Comments on the Quality of English Language
Double-checking abbreviations is recommended
Author Response
The response to the statistic issue is unsatisfactory as the median follow-up time is 18 months, as reported in the manuscript. I suggest trying a Cox regression instead of a logistic regression.
Thanks for this smart comment. Among the MACCE groups, most of the adverse events happened 3 to 6 months after the CMR examination. We consulted with statistics, and regarding no meaningful differences in time to event in the MACCEs group, analysis of the CMR variables with the Cox regression model was impossible. In case you recommend we can omit the logistic regression analysis.
Finally, I recommend enhancing the discussion/limitation section by delving deeper into the GLS and LGE issues that were previously reported. It would be beneficial to expand the bibliography by including recent and relevant research articles that specifically address these topics. This will help provide a more robust and up-to-date foundation for readers.
Thanks for your great advice. We comprehensively changed the discussion section and changes are marked with track change.
Round 3
Reviewer 2 Report
Comments and Suggestions for Authors
I appreciate the authors for their response. I still have some uncertainties regarding logistic regression.
I suggest including in the Result Section the average and median follow-up time between the group of patients who experienced a MAACE and those who did not. If there is a significant difference in time, I recommend performing a Cox regression or excluding the logistic multivariate analysis. This is a crucial point to clarify for the entire study.
Comments on the Quality of English LanguageMinor editing of the English language required
Author Response
I suggest including in the Result Section the average and median follow-up time between the group of patients who experienced a MAACE and those who did not. If there is a significant difference in time, I recommend performing a Cox regression or excluding the logistic multivariate analysis. This is a crucial point to clarify for the entire study.
Thank you very much for this great comment. As you suggested we omitted the logistic multivariate regression analysis from the manuscript to further clarify our work.
Round 4
Reviewer 2 Report
Comments and Suggestions for Authors
Thank you for your reply. Finally, the bibliography should be updated (in particular for the limitation section), as also suggested in the previous review.
Comments on the Quality of English LanguageMinor English revision.
Author Response
Thank you for your reply. Finally, the bibliography should be updated (in particular for the limitation section), as also suggested in the previous review.
Thanks for this comment. The bibliography section was revised especially for the "Limitation". Also, English editing was done.